# COUNTERFACTUAL GRAPH LEARNING FOR LINK PREDICTION

## ABSTRACT

Learning to predict missing links is important for many graph-based applications. Existing methods were designed to learn the association between two sets of variables: (1) the observed graph structure (e.g., clustering effect) and (2) the existence of link between a pair of nodes. However, the causal relationship between these variables was ignored. We visit the possibility of learning it by asking a counterfactual question: *"would the link exist or not if the observed graph structure became different?"* To answer this question, we leverage causal models considering the information of the node pair (i.e., learned graph representations) as context, global graph structural properties as treatment, and link existence as outcome. In this work, we propose a novel link prediction method that enhances graph learning by counterfactual inference. It creates counterfactual links from the observed ones, and learns representations from both the observed and counterfactual links. Experiments on benchmark datasets show that this novel graph learning method achieves state-of-the-art performance on link prediction.

## 1 INTRODUCTION

Link prediction seeks to predict the likelihood of edge existence between node pairs based on the observed graph. Given the omnipresence of graph-structured data, link prediction has copious applications such as movie recommendation (Bennett et al., 2007), chemical interaction prediction (Stanfield et al., 2017), and knowledge graph completion (Kazemi & Poole, 2018). Graph machine learning methods have been widely applied to solve this problem. Their standard scheme is to first learn the representation vectors of nodes and then learn the *association* between the representations of a pair of nodes and the existence of the link between them. For example, graph neural networks (GNNs) use neighborhood aggregation to create the representation vectors: the representation vector of a node is computed by recursively aggregating and transforming representation vectors of its neighboring nodes (Kipf & Welling, 2016a; Hamilton et al., 2017; Wu et al., 2020). Then the vectors are fed into a binary classification model to learn the *association*. GNN methods have shown predominance in the task of link prediction (Kipf & Welling, 2016b; Zhang et al., 2020a).

Unfortunately, the causal relationship between graph structure and link existence was largely ignored in previous work. Existing methods that learn from association ~~only were~~ are not able to capture essential factors to accurately predict missing links in the *test data*. Take social network as an example. Suppose Alice and Adam live in the same neighborhood and they are close friends. The association between neighborhood belonging and friend closeness could be too strong to discover the essential factors of the friendship such as common interests or family relationship which could be the cause of being living in the same neighborhood. So, our idea is asking a *counterfactual* question: *"would Alice and Adam still be close friends if they were not living in the same neighborhood?"* If a graph learning model could learn the causal relationship by answering the counterfactual questions, it would improve the accuracy of link prediction with the novel knowledge it captured. Generally, the questions can be described as *"would the link exist or not if the graph structure became different?"*

As known to many, counterfactual question is a key component of causal inference and have been well defined in the literature. A counterfactual question is usually framed with three factors: context (as a data point), manipulation (e.g., treatment, intervention, action, strategy), and outcome (van der Laan & Petersen, 2007; Johansson et al., 2016). (To simplify the language, we use "treatment" to refer to the manipulation in this paper, as readers might be familiar more with the word "treatment.")

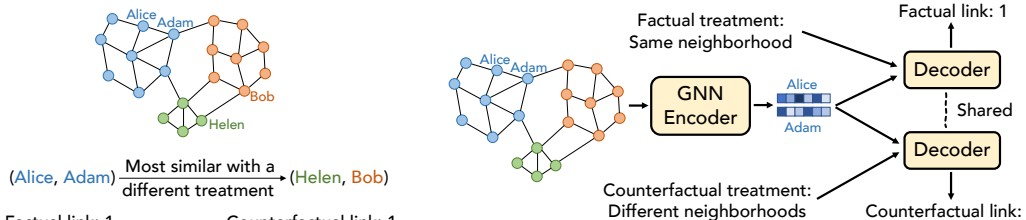

(a) Find counterfactual link as the most similar node pair with a different treatment.

(b) Train a GNN-based link predictor to predict factual and counterfactual links given the corresponding treatments.

Figure 1: The proposed CFLP learns the causal relationship between the observed graph structure (e.g., neighborhood similarity, considered as treatment variable) and link existence (considered as outcome). In this example, the link predictor would be trained to estimate the individual treatment effect (ITE) as $1 - 1 = 0$ so it looks for factors other than neighborhood to predict the factual link.

Given certain data context, it asks what the outcome would have been if the treatment had not been the observed value. In the scenario of link prediction, we consider the information of a pair of nodes as context, graph structural properties as treatment, and link existence as outcome. Recall the social network example. The context is the representations of Alice and Adam that are learned from their personal attributes and relationships with others on the network. The treatment is living in the same neighborhood, which can be identified by community detection. And the outcome is their friendship. In this work, we present a **c**ounter**f**actual graph learning method for **l**ink **p**rediction (CFLP) that trains graph learning models to answer the counterfactual questions. Figure 1 illustrates this two-step method. Suppose the treatment variable is defined as one type of global graph structure, e.g., the neighborhood assignment discovered by spectral clustering or community detection algorithms. We are wondering how likely the neighborhood distribution makes a difference on the link (non-)existence for each pair of nodes. So, given a pair of nodes (like Alice and Adam) and the treatment value on this pair (in the same neighborhood), we find a pair of nodes (like Helen and Bob) that satisfies two conditions: (1) it has a different treatment (in different neighborhoods) and (2) it is the most similar pair with the given pair of nodes. We call these matched pair of nodes as "counterfactual links." Note that the outcome of the counterfactual link can be either 1 or 0, depending on whether there exists an edge between the matched pair of nodes (Helen and Bob). The counterfactual link provides unobserved outcome to the given pair of nodes (Alice and Adam) under a counterfactual condition (in different neighborhoods). After counterfactual links are created for all (positive and negative) training examples, CFLP trains a link predictor (which can be GNN-based) to learn the representation vectors of nodes to predict both the observed factual links and the counterfactual links. In this Alice-Adam example, the link predictor is trained to estimate the individual treatment effect (ITE) of neighborhood assignment as $1 - 1 = 0$, where ITE is a metric for the effect of treatment on the outcome and zero indicates the given treatment has no effect on the outcome. So, the learner will try to discover the essential factors on the friendship between Alice and Adam. CFLP leverages causal models to find these factors for graph learning models to accurately predict missing links.

**Contributions.** Our main contributions can be summarized as follows. (1) This is the first work that proposes to improve link prediction by causal inference, specifically, learning to answer counterfactual questions about link existence. (2) This work introduces CFLP that trains GNN-based link predictors to predict both factual and counterfactual links. It learns the causal relationship between global graph structure and link existence. (3) CFLP outperforms competitive baseline methods on several benchmark datasets. We analyze the impact of counterfactual links as well as the choice of treatment variable. This work sheds insights for improving graph machine learning with causal analysis, which has not been extensively studied yet, when the other direction (machine learning for causal inference) has been studied for a long time.

## 2 PROBLEM DEFINITION

**Notations** Let $G = (\mathcal{V}, \mathcal{E})$ be an undirected graph of $N$ nodes, where $\mathcal{V} = \{v_1, v_2, \ldots, v_N\}$ is the set of nodes and $\mathcal{E} \subseteq \mathcal{V} \times \mathcal{V}$ is the set of observed links. We denote the adjacency matrix as

$\mathbf{A} \in \{0,1\}^{N \times N}$, where $A_{i,j} = 1$ indicates nodes $v_i$ and $v_j$ are connected and vice versa. We denote the node feature matrix as $\mathbf{X} \in \mathbb{R}^{N \times F}$, where $F$ is the number of node features and $\mathbf{x}_i$ (bolded) indicates the feature vector of node $v_i$ (the $i$-th row of $\mathbf{X}$).

In this work, we follow the commonly accepted problem definition of link prediction on graph data (Zhang & Chen, 2018; Zhang et al., 2020a; Cai et al., 2021): Given an observed graph $G$ (with validation and testing links masked off), predict the link existence between every pair of nodes. More specifically, for the GNN-based link prediction methods, they learn low-dimensional node representations $\mathbf{Z} \in \mathbb{R}^{N \times H}$, where $H$ is the dimensional size of latent space such that $H \ll F$, and then use $\mathbf{Z}$ for the prediction of link existence.

## 3 PROPOSED METHOD

### 3.1 IMPROVING GRAPH LEARNING WITH CAUSAL MODEL

**Leveraging Causal Model(s)** Counterfactual causal infer-ence aims to find out the causal relationship between treat-ment and outcomes by asking the counterfactual questions such as "would the outcome be different if the treatment was different?" (Morgan & Winship, 2015). Figure 2 is a typ-ical example, in which we denote the context (confounder) as $\mathbf{Z}$, treatment as $T$, and the outcome as $Y$. Given the context, treatments, and their corresponding outcomes, coun-terfactual inference methods aim to find the effect of treat-ment on the outcome, which is usually measured by *individ-ual treatment effect* (ITE) and its expectation *averaged treat-ment effect* (ATE) (van der Laan & Petersen, 2007; Weiss et al., 2015). For a binary treatment variable $T = \{0,1\}$, denoting $g(\mathbf{z}, T)$ as the outcome of $\mathbf{z}$ given the treatment $T$, we have $\text{ITE}(\mathbf{z}) = g(\mathbf{z}, 1) - g(\mathbf{z}, 0)$, and $\text{ATE} = \mathbb{E}_{\mathbf{z} \sim \mathbf{Z}} \, \text{ITE}(\mathbf{z})$.

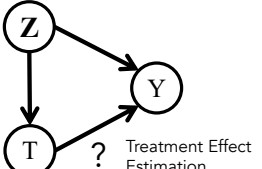

Figure 2: Causal modeling (not the target of our work but related to the idea we propose): Given $\mathbf{Z}$ and ob-served outcomes, find treatment ef-fect of $T$ on $Y$.

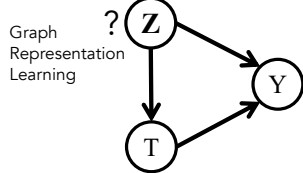

Figure 3: Graph learning with causal model (the proposed idea): leverage the estimated $\text{ATE}(Y|T)$ to improve the learning of $\mathbf{Z}$.

Ideally, we need all possible outcomes of the contexts under all kinds of treatments to study the causal relationships (Mor-gan & Winship, 2015). However, in reality, the fact that we can only observe one potential outcome under one particu-lar treatment prevents the ITE from being known (Johansson et al., 2016). Traditional causal inference methods use statisti-cal learning approaches such as Neyman–Rubin casual model (BCM) and propensity score matching (PSM) to predict the value of ATE (Rubin, 1974; 2005).

In this work, we look at *link prediction* with graph learning, which is essentially learning the best node representations $\mathbf{Z}$ for the prediction of link existence. Therefore, as shown in Figure 3, where the outcome $Y$ is the link existence, the objective is different from classic causal inference. In graph learning, we can estimate the effect of treatment on the outcome ($\text{ATE}(Y|T)$), and we want to improve the learning of $\mathbf{Z}$ with the estimation. More specifically, in graph learning for link prediction, for each pair of nodes $(v_i, v_j)$, its ITE can be estimated with

$$\text{ITE}_{(v_i, v_j)} = g((\mathbf{z}_i, \mathbf{z}_j), 1) - g((\mathbf{z}_i, \mathbf{z}_j), 0) \tag{1}$$

and we use this information to improve the learning of $\mathbf{Z}$, i.e., $P(\mathbf{Z}|Y)$.

We denote the observed adjacency matrix as the *factual* outcomes $\mathbf{A}$ and the unobserved adja-cency matrix when the treatment is different as the *counterfactual* outcomes $\mathbf{A}^{CF}$. We denote $\mathbf{T} \in \{0,1\}^{N \times N}$ as the binary factual treatment matrix, where $T_{i,j}$ indicates the treatment of the node pair $(v_i, v_j)$. We denote $\mathbf{T}^{CF}$ as the counterfactual treatment matrix where $T_{i,j}^{CF} = 1 - T_{i,j}$. We are interested in (1) estimating the counterfactual outcomes $\mathbf{A}^{CF}$ via observed data, (2) learn-ing with the counterfactual adjacency matrix $\mathbf{A}^{CF}$ to enhance link prediction, and (3) learning the causal relationship between graph structural information (treatment) and link existence (outcome).

**Treatment Variable** Previous work on graph machine learning (Velickovic et al., 2019; Park et al., 2020) showed that the graph's global structural information could improve the quality of representa-tion vectors of nodes learned by GNNs. This is because the message passing-based GNNs aggregate

local information in the algorithm of representation vector generation and the global structural information is complementary with the aggregated information. Therefore, for a pair of nodes, one option of defining the treatment variable is its global structural role in the graph. Without the loss of generality, we use Louvain (Blondel et al., 2008), an unsupervised approach that has been widely used for community detection, as an example. Louvain discovers community structure of a graph and assigns each node to one community. Then we can define the binary treatment variable as whether these two nodes in the pair belong to the same community. Let $c : \mathcal{V} \to \mathbb{N}$ be any graph mining/clustering method that outputs the index of community/cluster/neighborhood that each node belongs to. The treatment matrix $\mathbf{T}$ is defined as $T_{i,j} = 1$ if $c(v_i) = c(v_j)$, and $T_{i,j} = 0$ otherwise. For the choice of $c$, we suggest methods that group nodes based on global graph structural information, including but not limited to Louvain (Blondel et al., 2008), K-core (Bader & Hogue, 2003), and spectral clustering (Ng et al., 2001).

## 3.2 COUNTERFACTUAL LINKS

To implement the solution based on above idea, we propose counterfactual links. As aforementioned, for each node pair, the observed data contains only the factual treatment and outcome, meaning that the link existence for the given node pair with an opposite treatment is unknown. Therefore, we use the outcome from the nearest observed context as a substitute. This type of matching on covariates is widely used to estimate treatment effects from observational data (Johansson et al., 2016; Alaa & Van Der Schaar, 2019). That is, we want to find the nearest neighbor with the opposite treatment for each observed node pairs and use the nearest neighbor's outcome as a *counterfactual link*. Formally, $\forall (v_i, v_j) \in \mathcal{V} \times \mathcal{V}$, we want to find its counterfactual link $(v_a, v_b)$ as below:

$$(v_a, v_b) = \underset{v_a, v_b \in \mathcal{V}}{\arg\min} \{ h((v_i, v_j), (v_a, v_b)) \mid T_{a,b} = 1 - T_{i,j} \}, \tag{2}$$

where $h(\cdot, \cdot)$ is a metric of measuring the distance between a pair of node pairs (a pair of contexts). Nevertheless, finding the nearest neighbors by computing the distance between all pairs of node pairs is extremely inefficient and infeasible in application, which takes $O(N^4)$ comparisons (as there are totally $O(N^2)$ node pairs). Hence we implement Eq. (2) using node-level embeddings. Specifically, considering that we want to find the nearest node pair based on not only the raw node features but also structural features, we take the state-of-the-art unsupervised graph representation learning method MVGRL (Hassani & Khasahmadi, 2020) to learn the node embeddings $\tilde{\mathbf{X}} \in \mathbb{R}^{N \times \tilde{F}}$ from the observed graph (with validation and testing links masked off). We use $\tilde{\mathbf{X}}$ to find the nearest neighbors of node pairs. Therefore, $\forall (v_i, v_j) \in \mathcal{V} \times \mathcal{V}$, we define its counterfactual link $(v_a, v_b)$ as

$$(v_a, v_b) = \underset{v_a, v_b \in \mathcal{V}}{\arg\min} \{ d(\tilde{\mathbf{x}}_i, \tilde{\mathbf{x}}_a) + d(\tilde{\mathbf{x}}_j, \tilde{\mathbf{x}}_b) \mid T_{a,b} = 1 - T_{i,j}, d(\tilde{\mathbf{x}}_i, \tilde{\mathbf{x}}_a) + d(\tilde{\mathbf{x}}_j, \tilde{\mathbf{x}}_b) < 2\gamma \}, \tag{3}$$

where $d(\cdot, \cdot)$ is specified as the Euclidean distance on the embedding space of $\tilde{\mathbf{X}}$, and $\gamma$ is a hyperparameter that defines the maximum distance that two nodes are considered as similar. When no node pair satisfies the above equation (i.e., there does not exist any node pair with opposite treatment that is close enough to the target node pair), we do not assign any nearest neighbor for the given node pair to ensure all the neighbors are similar enough (as substitutes) in the feature space. Thus, the counterfactual treatment matrix $\mathbf{T}^{CF}$ and the counterfactual adjacency matrix $\mathbf{A}^{CF}$ are defined as

$$T_{i,j}^{CF}, A_{i,j}^{CF} = \begin{cases} 1 - T_{i,j}, A_{a,b} & \text{, if } \exists\, (v_a, v_b) \in \mathcal{V} \times \mathcal{V} \text{ satisfies Eq. (3);} \\ T_{i,j}, A_{i,j} & \text{, otherwise.} \end{cases} \tag{4}$$

It is worth noting that the node embeddings $\tilde{\mathbf{X}}$ and the nearest neighbors are computed only once and do not change during the learning process. $\tilde{\mathbf{X}}$ is only used for finding the nearest neighbors. We also note that $\tilde{\mathbf{X}}$ must be structural embeddings rather than positional embeddings (as defined in (Srinivasan & Ribeiro, 2020)).

**Learning from Counterfactual Distributions** Let $P^F$ be the factual distribution of the observed contexts and treatments, and $P^{CF}$ be the counterfactual distribution that is composed of the observed contexts and opposite treatments. We define the empirical factual distribution $\hat{P}^F \sim P^F$ as $\hat{P}^F = \{(v_i, v_j, T_{i,j})\}_{i,j=1}^N$, and define the empirical counterfactual distribution $\hat{P}^{CF} \sim P^{CF}$ as $\hat{P}^{CF} = \{(v_i, v_j, T_{i,j}^{CF})\}_{i,j=1}^N$. Unlike traditional link prediction methods that take only $\hat{P}^F$ as input and use the observed outcomes $\mathbf{A}$ as the training target, the idea of counterfactual graph learning is to take advantage of the counterfactual distribution by having $\hat{P}^{CF}$ as a complementary input and use the counterfactual outcomes $\mathbf{A}^{CF}$ as the training target for the counterfactual data samples.

### 3.3 THE COUNTERFACTUAL GRAPH LEARNING MODEL

In this subsection, we present the design of our model as well as the training method. The input of the model in CFLP includes (1) the observed graph data $\mathbf{A}$ and raw feature matrix $\mathbf{X}$, (2) the factual treatments $\mathbf{T}^F$ and counterfactual treatments $\mathbf{T}^{CF}$, and (3) the counterfactual graph data $\mathbf{A}^{CF}$. The output contains link prediction logits in $\widehat{\mathbf{A}}$ and $\widehat{\mathbf{A}}^{CF}$ for the factual and counterfactual adjacency matrices $\mathbf{A}$ and $\mathbf{A}^{CF}$, respectively.

**Graph Learning Model**  The model consist of two trainable components: a graph encoder $f$ and a link decoder $g$. The graph encoder generates representation vectors of nodes from graph data $G$. And the link decoder projects the representation vectors of node pairs into the link prediction logits. The choice of the graph encoder $f$ can be any end-to-end GNN model. Without the loss of generality, here we use the commonly used graph convolutional network (GCN) (Kipf & Welling, 2016a). Each layer of GCN is defined as

$$\mathbf{H}^{(l)} = f^{(l)}(\mathbf{A}, \mathbf{H}^{(l-1)}; \mathbf{W}^{(l)}) = \sigma(\tilde{\mathbf{D}}^{-\frac{1}{2}}\tilde{\mathbf{A}}\tilde{\mathbf{D}}^{-\frac{1}{2}}\mathbf{H}^{(l-1)}\mathbf{W}^{(l)}), \tag{5}$$

where $l$ is the layer index, $\tilde{\mathbf{A}} = \mathbf{A} + \mathbf{I}$ is the adjacency matrix with added self-loops, $\tilde{\mathbf{D}}$ is the diagonal degree matrix $\tilde{D}_{ii} = \sum_j \tilde{A}_{ij}$, $\mathbf{H}^{(0)} = \mathbf{X}$, $\mathbf{W}^{(l)}$ is the learnable weight matrix at the $l$-th layer, and $\sigma(\cdot)$ denotes a nonlinear activation such as ReLU. We denote $\mathbf{Z} = f(\mathbf{A}, \mathbf{X}) \in \mathbb{R}^{N \times H}$ as the output from the encoder's last layer, i.e., the $H$-dimensional representation vectors of nodes. Following previous work (Zhang et al., 2020a), we compute the representation of a node pair as the Hadamard product of the vectors of the two nodes. That is, the representation for the node pair $(v_i, v_j)$ is $\mathbf{z}_i \odot \mathbf{z}_j \in \mathbb{R}^H$, where $\odot$ stands for the Hadamard product.

For the link decoder that predicts whether a link exists between a pair of nodes, we opt for simplicity and adopt a simple decoder based on multi-layer perceptron (MLP), given the representations of node pairs and their treatments. That is, the decoder $g$ is defined as

$$\widehat{\mathbf{A}} = g(\mathbf{Z}, \mathbf{T}), \text{ where } \widehat{A}_{i,j} = \text{MLP}([\mathbf{z}_i \odot \mathbf{z}_j, T_{i,j}]), \tag{6}$$

$$\widehat{\mathbf{A}}^{CF} = g(\mathbf{Z}, \mathbf{T}^{CF}), \text{ where } \widehat{A}_{i,j}^{CF} = \text{MLP}([\mathbf{z}_i \odot \mathbf{z}_j, T_{i,j}^{CF}]), \tag{7}$$

where $[\cdot, \cdot]$ stands for the concatenation of vectors, and $\widehat{\mathbf{A}}$ and $\widehat{\mathbf{A}}^{CF}$ can be used for estimating the observed ITE as aforementioned in Eq. (1).

During the training process, data samples from the empirical factual distribution $\hat{P}^F$ and the empirical counterfactual distribution $\hat{P}^{CF}$ are fed into decoder $g$ and optimized towards $\mathbf{A}$ and $\mathbf{A}^{CF}$, respectively. That is, for the two distributions, the loss functions are as follows:

$$\mathcal{L}_F = \frac{1}{N^2} \sum_{i=1}^{N} \sum_{j=1}^{N} A_{i,j} \cdot \log \widehat{A}_{i,j} + (1 - A_{i,j}) \cdot \log(1 - \widehat{A}_{i,j}), \tag{8}$$

$$\mathcal{L}_{CF} = \frac{1}{N^2} \sum_{i=1}^{N} \sum_{j=1}^{N} A_{i,j}^{CF} \cdot \log \widehat{A}_{i,j}^{CF} + (1 - A_{i,j}^{CF}) \cdot \log(1 - \widehat{A}_{i,j}^{CF}). \tag{9}$$

**Balancing Counterfactual Learning**  In the training process, the above loss minimizations train the model on both the empirical factual distribution $\hat{P}^F \sim P^F$ and empirical counterfactual distribution $\hat{P}^{CF} \sim P^{CF}$ that are not necessarily equal – the training examples (node pairs) do not have to be aligned. However, at the stage of inference, the test data contains only observed (factual) samples. Such a gap between the training and testing data distributions exposes the model in the risk of covariant shift, which is a common issue in counterfactual learning (Johansson et al., 2016; Assaad et al., 2021).

To force the distributions of representations of factual distributions and counterfactual distributions to be similar, we use the discrepancy distance (Mansour et al., 2009; Johansson et al., 2016) as another objective to regularize the representation learning. That is, we use the following loss term to minimize the distance between the learned representations from $\hat{P}^F$ and $\hat{P}^{CF}$:

$$\mathcal{L}_{disc} = \text{disc}(\hat{P}_f^F, \hat{P}_f^{CF}), \text{ where } \text{disc}(P, Q) = ||P - Q||_F, \tag{10}$$

where $|| \cdot ||_F$ denotes the Frobenius Norm, and $\hat{P}_f^F$ and $\hat{P}_f^{CF}$ denote the node pair representations learned by graph encoder $f$ from factual distribution and counterfactual distribution, respectively. That is, the learned representations for $(v_i, v_j, T_{i,j})$ and $(v_i, v_j, T_{i,j}^{CF})$ are $[\mathbf{z}_i \odot \mathbf{z}_j, T_{i,j}]$ (Eq. (6)) and $[\mathbf{z}_i \odot \mathbf{z}_j, T_{i,j}^{CF}]$ (Eq. (7)), respectively.

**Training** During the training of CFLP, we want the model to be optimized towards three targets: (1) accurate link prediction on the observed outcomes (Eq. (8)), (2) accurate estimation on the counterfactual outcomes (Eq. (9)), and (3) regularization on the representation spaces learned from $\hat{P}^F$ and $\hat{P}^{CF}$ (Eq. (10)). Therefore, the overall training loss of our proposed CFLP is

$$\mathcal{L} = \mathcal{L}_F + \alpha \cdot \mathcal{L}_{CF} + \beta \cdot \mathcal{L}_{disc}, \tag{11}$$

where $\alpha$ and $\beta$ are hyperparameters to control the weights of counterfactual outcome estimation (link prediction) loss and discrepancy loss.

**Summary** Algorithm 1 summarizes the whole process of CFLP. The **first step** is to compute the factual and counterfactual treatments $\mathbf{T}$, $\mathbf{T}^{CF}$ as well as the counterfactual outcomes $\mathbf{A}^{CF}$. Then, the **second step** trains the graph learning model on both the observed factual data and created counterfactual data with the integrated loss function (Eq. (11)). Note that the discrepancy loss (Eq. (10)) is computed on the representations of node pairs learned by the graph encoder $f$, so the decoder $g$ is trained with data from both $\hat{P}^F$ and $\hat{P}^{CF}$ without balancing the constraints. Therefore, after the model is sufficiently trained, we freeze the graph encoder $f$ and fine-tune $g$ with only the factual data. Finally, after the decoder is sufficiently fine-tuned, we output the link prediction logits for both the factual and counterfactual adjacency matrices.

---

**Algorithm 1:** CFLP: Counterfactual graph learning for link prediction

**Input** : $f$, $g$, $\mathbf{A}$, $\mathbf{X}$, $n\_epochs$, $n\_epoch\_ft$

1 Compute $\mathbf{T}$ as presented in Section 3.1 ;
2 Compute $\mathbf{T}^{CF}$, $\mathbf{A}^{CF}$ by Eqs. (3) and (4) ;
   /* model training                    */
3 Initialize $\Theta_f$ in $f$ and $\Theta_g$ in $g$ ;
4 **for** *epoch in range(n_epochs)* **do**
5     $\mathbf{Z} = f(\mathbf{A}, \mathbf{X})$ ;
6     Get $\widehat{\mathbf{A}}$ and $\widehat{\mathbf{A}}^{CF}$ via $g$ with Eqs. (6) and (7) ;
7     Update $\Theta_f$ and $\Theta_g$ with $\mathcal{L}$ ;    // Eq. (11)
8 **end**
   /* decoder fine-tuning              */
9 Freeze $\Theta_f$ and re-initialize $\Theta_g$ ;
10 $\mathbf{Z} = f(\mathbf{A}, \mathbf{X})$ ;
11 **for** *epoch in range(n_epochs_ft)* **do**
12     Get $\widehat{\mathbf{A}}$ via $g$ with Eq. (6) ;
13     Update $\Theta_g$ with $\mathcal{L}_F$ ;      // Eq. (8)
14 **end**
   /* ~~model inferencing~~ inference    */
15 $\mathbf{Z} = f(\mathbf{A}, \mathbf{X})$ ;
16 Get $\widehat{\mathbf{A}}$ and $\widehat{\mathbf{A}}^{CF}$ via $g$ with Eqs. (6) and (7) ;
**Output:** $\widehat{\mathbf{A}}$ for link prediction, $\widehat{\mathbf{A}}^{CF}$

---

**Complexity** The complexity of the first step (finding counterfactual links with nearest neighbors) is proportional to the number of node pairs. When $\gamma$ is set as a small value to obtain indeed similar node pairs, this step (Eq. (3)) uses constant time. Moreover, the computation in Eq. (3) can be parallelized. Therefore, the time complexity is $O(N^2/C)$ where $C$ is the number of processes. For the complexity of the second step (training counterfactual learning model), the GNN encoder has time complexity of $O(LH^2N + LH|\mathcal{E}|)$ (Wu et al., 2020), where $L$ is the number of GNN layers and $H$ is the size of node representations. Given that we sample the same number of non-existing links as that of observed links during training, the complexity of a *three-layer MLP* decoder is $O(((H + 1) \cdot d_h + d_h \cdot 1)|\mathcal{E}|) = O(d_h(H + 2)|\mathcal{E}|)$, where $d_h$ is the number of neurons in the hidden layer. Therefore, the second step has linear time complexity w.r.t. the sum of node and edge counts.

**Limitations** First, as mentioned above, the computation of finding counterfactual links has a worst-case complexity of $O(N^2)$. Second, CFLP performs counterfactual prediction with only a single treatment; however, there are quite a few kinds of graph structural information that can be considered as treatments. Future work can leverage the rich structural information by bundled treatments (Zou et al., 2020) in counterfactual graph learning.

Table 1: Link prediction performances measured by Hits@20. Best performance and best baseline performance are marked with bold and underline, respectively.

| | CORA | CITESEER | PUBMED | FACEBOOK | OGB-DDI |
|---|---|---|---|---|---|
| Node2Vec | 49.96±2.51 | 47.78±1.72 | 39.19±1.02 | 24.24±3.02 | 23.26±2.09 |
| MVGRL | 19.53±2.64 | 14.07±0.79 | 14.19±0.85 | 14.43±0.33 | 10.02±1.01 |
| VGAE | 45.91±3.38 | 44.04±4.86 | 23.73±1.61 | 37.01±0.63 | 11.71±1.96 |
| SEAL | 51.35±2.26 | 40.90±3.68 | 28.45±3.81 | 40.89±5.70 | 30.56±3.86 |
| LGLP | 62.98±0.56 | 57.43±3.71 | – | 37.86±2.13 | – |
| GCN | 49.06±1.72 | 55.56±1.32 | 21.84±3.87 | 53.89±2.14 | 37.07±5.07 |
| GSAGE | 53.54±2.96 | 53.67±2.94 | 39.13±4.41 | 45.51±3.22 | 53.90±4.74 |
| JKNet | 48.21±3.86 | 55.60±2.17 | 25.64±4.11 | 52.25±1.48 | 60.56±8.69 |
| Our proposed CFLP with different graph encoders | | | | | |
| CFLP w/ GCN | 60.34±2.33 | 59.45±2.30 | 34.12±2.72 | 53.95±2.29 | 52.51±1.09 |
| CFLP w/ GSAGE | 57.33±1.73 | 53.05±2.07 | 43.07±2.36 | 47.28±3.00 | 75.49±4.33 |
| CFLP w/ JKNet | **65.57**±1.05 | **68.09**±1.49 | **44.90**±2.00 | **55.22**±1.29 | **86.08**±1.98 |

## 4 EXPERIMENTS

### 4.1 EXPERIMENTAL SETUP

We conduct experiments on five benchmark datasets including citation networks (CORA, CITE-SEER, PUBMED (Yang et al., 2016)), social network (FACEBOOK (McAuley & Leskovec, 2012)), and drug-drug interaction network (OGB-DDI (Wishart et al., 2018)) from the Open Graph Benchmark (OGB) (Hu et al., 2020). For the first four datasets, we randomly select 10%/20% of the links and the same numbers of disconnected node pairs as validation/test samples. The links in the validation and test sets are masked off from the training graph. For OGB-DDI, we used the OGB official train/validation/test splits. Statistics and details for the datasets are given in Appendix. We use K-core (Bader & Hogue, 2003) clusters as the default treatment variable. We evaluate CFLP on three commonly used GNN encoders: GCN (Kipf & Welling, 2016a), GSAGE (Hamilton et al., 2017), and JKNet (Xu et al., 2018). We compare the link prediction performance of CFLP against Node2Vec (Grover & Leskovec, 2016), MVGRL (Hassani & Khasahmadi, 2020), VGAE (Kipf & Welling, 2016b), SEAL (Zhang & Chen, 2018), LGLP (Cai et al., 2021), and GNNs with MLP decoder. We report averaged test performance and their standard deviation over 20 runs with different random parameter initializations. Other than the most commonly used of Area Under ROC Curve (AUC), we report Hits@20 (one of the primary metrics on OGB leaderboard) as a more challenging metric, as it expects models to rank positive edges higher than nearly all negative edges.

Besides performance comparison on link prediction, we will answer two questions to suggest a way of choosing a treatment variable for creating counterfactual links: (Q1) Does CFLP sufficiently learn the observed *averaged treatment effect* (ATE) derived from the counterfactual links? (Q2) What is the relationship between the estimated ATE learned in the method and the prediction performance? If the answer to Q1 is yes, then the answer to Q2 will indicate how to choose treatment based on observed ATE. To answer the Q1, we calculate the observed ATE ($\widehat{\text{ATE}}_{obs}$) by comparing the observed links in $\mathbf{A}$ and created counterfactual links $\mathbf{A}^{CF}$ that have opposite treatments. And we calculate the estimated ATE ($\widehat{\text{ATE}}_{est}$) by comparing the predicted links in $\widehat{\mathbf{A}}$ and predicted counterfactual links $\widehat{\mathbf{A}}^{CF}$. Formally, $\widehat{\text{ATE}}_{obs}$ and $\widehat{\text{ATE}}_{est}$ are defined as

$$\widehat{\text{ATE}}_{obs} = \frac{1}{N^2} \sum_{i=1}^{N} \sum_{j=1}^{N} \left\{ \mathbf{T} \odot (\mathbf{A} - \mathbf{A}^{CF}) + (\mathbf{1}_{N \times N} - \mathbf{T}) \odot (\mathbf{A}^{CF} - \mathbf{A}) \right\}_{i,j}. \quad (12)$$

$$\widehat{\text{ATE}}_{est} = \frac{1}{N^2} \sum_{i=1}^{N} \sum_{j=1}^{N} \left\{ \mathbf{T} \odot (\widehat{\mathbf{A}} - \widehat{\mathbf{A}}^{CF}) + (\mathbf{1}_{N \times N} - \mathbf{T}) \odot (\widehat{\mathbf{A}}^{CF} - \widehat{\mathbf{A}}) \right\}_{i,j}. \quad (13)$$

The treatment variables we will investigate are usually graph clustering or community detection methods, such as K-core (Bader & Hogue, 2003), stochastic block model (SBM) (Karrer & Newman, 2011), spectral clustering (SpecC) (Ng et al., 2001), propagation clustering (PropC) (Raghavan et al., 2007), Louvain (Blondel et al., 2008), common neighbors (CommN), Katz index, and hierarchical clustering (Ward) (Ward Jr, 1963). We use JKNet (Xu et al., 2018) as default graph encoder.

Table 2: Link prediction performances measured by AUC. Best performance and best baseline performance are marked with bold and underline, respectively.

| | CORA | CITESEER | PUBMED | FACEBOOK | OGB-DDI |
|---|---|---|---|---|---|
| Node2Vec | 84.49±0.49 | 80.00±0.68 | 80.32±0.29 | 86.49±4.32 | 90.83±0.02 |
| MVGRL | 75.07±3.63 | 61.20±0.55 | 80.78±1.28 | 79.83±0.30 | 81.45±0.99 |
| VGAE | 88.68±0.40 | 85.35±0.60 | 95.80±0.13 | 98.66±0.04 | 93.08±0.15 |
| SEAL | 92.55±0.50 | 85.82±0.44 | 96.36±0.28 | **99.60**±0.02 | 97.85±0.17 |
| LGLP | 91.30±0.05 | 89.41±0.13 | – | 98.51±0.01 | – |
| GCN | 90.25±0.53 | 71.47±1.40 | 96.33±0.80 | 99.43±0.02 | 99.82±0.05 |
| GSAGE | 90.24±0.34 | 87.38±1.39 | 96.78±0.11 | 99.29±0.04 | 99.93±0.02 |
| JKNet | 89.05±0.67 | 88.58±1.78 | 96.58±0.23 | 99.43±0.02 | **99.94**±0.01 |
| Our proposed CFLP with different graph encoders | | | | | |
| CFLP w/ GCN | 92.55±0.50 | 89.65±0.20 | 96.99±0.08 | 99.38±0.01 | 99.44±0.05 |
| CFLP w/ GSAGE | 92.61±0.52 | 91.84±0.20 | 97.01±0.01 | 99.34±0.10 | 99.83±0.05 |
| CFLP w/ JKNet | **93.05**±0.24 | **92.12**±0.47 | **97.53**±0.17 | 99.31±0.04 | **99.94**±0.01 |

Table 3: Results of CFLP with different treatments on CORA. (sorted by Hits@20)

| | Hits@20 | $\widehat{\text{ATE}}_{obs}$ | $\widehat{\text{ATE}}_{est}$ |
|---|---|---|---|
| K-core | 65.6±1.1 | 0.002 | 0.013±0.003 |
| SBM | 64.2±1.1 | 0.006 | 0.023±0.015 |
| CommN | 62.3±1.6 | 0.007 | 0.053±0.021 |
| PropC | 61.7±1.4 | 0.037 | 0.059±0.065 |
| Ward | 61.2±2.3 | 0.001 | 0.033±0.012 |
| SpecC | 59.3±2.8 | 0.002 | 0.033±0.011 |
| Louvain | 57.6±1.8 | 0.025 | 0.138±0.091 |
| Katz | 56.6±3.4 | 0.740 | 0.802±0.041 |

Table 4: Results of CFLP with different treatments on CITESEER. (sorted by Hits@20)

| | Hits@20 | $\widehat{\text{ATE}}_{obs}$ | $\widehat{\text{ATE}}_{est}$ |
|---|---|---|---|
| SBM | 71.6±1.9 | 0.004 | 0.005±0.001 |
| K-core | 68.1±1.5 | 0.002 | 0.010±0.002 |
| Ward | 67.0±1.7 | 0.003 | 0.037±0.009 |
| PropC | 64.6±3.6 | 0.141 | 0.232±0.113 |
| Louvain | 63.3±2.5 | 0.126 | 0.151±0.078 |
| SpecC | 59.9±1.3 | 0.009 | 0.166±0.034 |
| Katz | 57.3±0.5 | 0.245 | 0.224±0.037 |
| CommN | 56.8±4.9 | 0.678 | 0.195±0.034 |

Implementation details and supplementary experimental results (e.g., sensitivity on $\gamma$, ablation study on $\mathcal{L}_{CF}$ and $\mathcal{L}_{disc}$) can be found in Appendix. Source code is available in supplementary material.

## 4.2 EXPERIMENTAL RESULTS

**Link Prediction** Tables 1 and 2 show the link prediction performance of Hits@20 and AUC by all methods. LGLP on PUBMED and OGB-DDI are missing due to the out of memory error when running the code package from the authors. We observe that our CFLP on different graph encoders achieve similar or better performances compared with baselines. The only exception is the AUC on FACEBOOK where most methods have close-to-perfect AUC. As AUC is a relatively easier metric comparing with Hits@20, most methods achieved good performance on AUC. We observe that CFLP with JKNet almost consistently achieves the best performance and outperforms baselines significantly on Hits@20. Specifically, comparing with the best baseline, CFLP improves relatively by 16.4% and 0.8% on Hits@20 and AUC, respectively. Comparing with the best performing baselines, which are also GNN-based, CFLP benefits from learning with both observed link existence ($\mathbf{A}$) and our defined counterfactual links ($\mathbf{A}^{CF}$).

**ATE with Different Treatments** Tables 3 and 4 show the link prediction performance, $\widehat{\text{ATE}}_{obs}$, and $\widehat{\text{ATE}}_{est}$ of CFLP (with JKNet) when using different treatments. The treatments in Tables 3 and 4 are sorted by the Hits@20 performance. Bigger ATE indicates stronger causal relationship between the treatment and outcome, and vice versa. We observe: (1) the rankings of $\widehat{\text{ATE}}_{est}$ and $\widehat{\text{ATE}}_{obs}$ are positively correlated with Kendell's ranking coefficient (Abdi, 2007) of 0.67 and 0.57 for CORA and CITESEER, respectively. Hence, CFLP was sufficiently trained to learn the causal relationship between graph structure information and link existence; (2) $\widehat{\text{ATE}}_{obs}$ and $\widehat{\text{ATE}}_{est}$ are both negatively correlated with the link prediction performance, showing that we can pick a proper treatment prior to training a model with CFLP. Using the treatment that has the weakest causal relationship with link existence is likely to train the model to capture more essential factors on the outcome, in a way similar to denoising the unrelated information from the representations.

## 5    RELATED WORK

**Link Prediction**    With its wide applications, link prediction has draw attention from many research communities including statistical machine learning and data mining. Stochastic generative methods based on stochastic block models (SBM) are developed to generate links (Mehta et al., 2019). In data mining, matrix factorization (Menon & Elkan, 2011), heuristic methods (Philip et al., 2010; Martínez et al., 2016), and graph embedding methods (Cui et al., 2018) have been applied to predict links in the graph. Heuristic methods compute the similarity score of nodes based on their neighborhoods. These methods can be generally categorized into first-order, second-order, and high-order heuristics based on the maximum distance of the neighbors. Graph embedding methods learn latent node features via embedding lookup and use them for link prediction (Perozzi et al., 2014; Tang et al., 2015; Grover & Leskovec, 2016; Wang et al., 2016).

In the past few years, GNNs have showed promising results on various graph-based tasks with their ability of learning from features and custom aggregations on structures (Kipf & Welling, 2016a; Hamilton et al., 2017; Wu et al., 2020)(Cotta et al., 2021). With node pair representations and an attached MLP or inner-product decoder, GNNs can be used for link prediction (Zhang et al., 2020a; Davidson et al., 2018; Yang et al., 2018). For example, VGAE used GCN to learn node representations and reconstruct the graph structure (Kipf & Welling, 2016b). SEAL extracted a local subgraph around each target node pair and then learned graph representation from local subgraph for link prediction (Zhang & Chen, 2018). Following the scheme of SEAL, Cai & Ji (2020) proposed to improve local subgraph representation learning by multi-scale graph representation learning. And LGLP inverted the local subgraphs to line graphs before learning representations (Cai et al., 2021). However, very limited work has studied to use causal inference for improving link prediction.

**Counterfactual Prediction**    As a mean of learning the causality between treatment and outcome, counterfactual prediction has been used for a variety of applications such as recommender systems (Wang et al., 2020; Xu et al., 2020), health care (Alaa & van der Schaar, 2017; Pawlowski et al., 2020), vision-language tasks (Zhang et al., 2020b; Parvaneh et al., 2020), and decision making (Coston et al., 2020; Pitis et al., 2020; Kusner et al., 2017). To infer the causal relationships, previous work usually estimated the ITE via function fitting models (Gelman & Hill, 2006; Chipman et al., 2010; Wager & Athey, 2018; Assaad et al., 2021). Peysakhovich et al. (2019) and Zou et al. (2020) studied counterfactual prediction with multiple agents and bundled treatments, respectively.

**Causal Inference**    Causal inference methods usually re-weighted samples based on propensity score (Rosenbaum & Rubin, 1983; Austin, 2011) to remove confounding bias from binary treatments. Recently, several works studied about learning treatment invariant representation to predict the counterfactual outcomes (Shalit et al., 2017; Li & Fu, 2017; Yao et al., 2018; Yoon et al., 2018; Hassanpour & Greiner, 2019a;b; Bica et al., 2020). Few recent works combined causal inference with graph learning (Sherman & Shpitser, 2020; Bevilacqua et al., 2021; Lin et al., 2021; Feng et al., 2021). For example, Sherman & Shpitser (2020) proposed a new concept in causal modeling, called "network intervention", to study the effect of link creation on network structure changes. Bevilacqua et al. (2021) studied the task of out-of-distribution (OOD) graph classification, and showed how subgraph densities can be used to build size-invariant graph representations, which alleviates the train-test gap when learning from OOD data.

## 6    CONCLUSION AND FUTURE WORK

In this work, we presented a counterfactual graph learning method for link prediction (CFLP). We introduced the idea of counterfactual prediction to improve link prediction on graphs. CFLP accurately predicted the missing links by exploring the causal relationship between global graph structure and link existence. Extensive experiments demonstrated that CFLP achieved the state-of-the-art performance on benchmark datasets. This work sheds insights that a good use of causal models (even basic ones) can greatly improve the performance of (graph) machine learning tasks, which in our case is link prediction. We note that the use of more sophistically designed causal models may lead to larger improvements for other machine learning tasks, which could be a valuable future research direction for the community. Other than our use of global graph structure as treatment, other treatments choices (with both empirical and theoretical analyses) are also worth exploring. Moreover, as CFLP first generates counterfactual links and then learns from both observed and counterfactual link existence, the underlying philosophy of our methodology could be considered as graph data augmentation. Therefore, investigating the relationship between counterfactual graph learning and graph data augmentation is also a possible future research direction.

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

Table 5: Statistics of datasets used in the experiments.

| Dataset | CORA | CITESEER | PUBMED | FACEBOOK | OGB-DDI |
|---|---|---|---|---|---|
| # nodes | 2,708 | 3,327 | 19,717 | 4,039 | 4,267 |
| # links | 5,278 | 4,552 | 44,324 | 88,234 | 1,334,889 |
| # validation node pairs | 1,054 | 910 | 8,864 | 17,646 | 235,371 |
| # test node pairs | 2,110 | 1,820 | 17,728 | 35,292 | 229,088 |

# APPENDICES

## A  ADDITIONAL DATASET DETAILS

In this section, we provide some additional dataset details. All the datasets used in this work are publicly available. Statistics for the datasets are shown in Table 5.

**Citation Networks**  CORA, CITESEER, and PUBMED are citation networks that were first used by Yang et al. (2016) and then commonly used as benchmarks in GNN-related literature (Kipf & Welling, 2016a; Veličković et al., 2017). In these citation networks, the nodes are published papers and features are bag-of-word vectors extracted from the corresponding paper. Links represent the citation relation between papers. We loaded the datasets with the `DGL`[1] package.

**Social Network**  The FACEBOOK dataset[2] is a social network constructed from friends lists from Facebook (McAuley & Leskovec, 2012). The nodes are Facebook users and links indicate the friendship relation on Facebook. The node features were constructed from the user profiles and anonymized by McAuley & Leskovec (2012).

**Drug-Drug Interaction Network**  The OGB-DDI dataset was constructed from a public Drug database (Wishart et al., 2018) and provided by the Open Graph Benchmark (OGB) (Hu et al., 2020). Each node in this graph represents an FDA-approved or experimental drug and edges represent the existence of unexpected effect when the two drugs are taken together. This dataset does not contain any node features, and it can be downloaded with the dataloader[3] provided by OGB.

## B  EXPANDED RELATED WORK

With the rapid development of graph machine learning in the past few years, researchers have been attempting to relate graph neural networks (GNNs) with causal models. Recently, several works have been proposed to improve graph learning with causal models (Sherman & Shpitser, 2020; Bevilacqua et al., 2021; Lin et al., 2021; Feng et al., 2021). Sherman & Shpitser (2020) proposed a new concept in causal modeling, called "network intervention", that is a type of structural intervention in network contexts. Sherman & Shpitser (2020) modeled social network with causal DAG and studied the effect of network intervention (link creation and removal) on network structure changes. Lin et al. (2021) formulated the problem of post-hoc explanation generation for GNNs as a causal learning task and proposed a causal explanation model with a loss designed based on Granger causality. Feng et al. (2021) formulated node classification of GNNs with a causal DAG, which estimated the causal effect of the local structure on the prediction and adaptively chose whether to aggregate from the neighbors. Bevilacqua et al. (2021) studied the task of out-of-distribution (OOD) graph classification, and showed how subgraph densities can be used to build size-invariant graph representations. They modeled OOD graph classification with a twin network DAG causal model, which learned approximately environment-invariant graph representations that better extrapolate between train and test data. The last three works, i.e., Lin et al. (2021), Feng et al. (2021), Bevilacqua et al. (2021), proposed to use causal models to improve the performance of three different types of graph machine learning tasks such as GNN explanation (subgraph) generation, node-level classification,

---

[1] https://github.com/dmlc/dgl

[2] https://snap.stanford.edu/data/ego-Facebook.html

[3] https://ogb.stanford.edu/docs/linkprop/#data-loader

and graph-level classification. Compared with them, our work has three points of uniqueness. First, to the best of our knowledge, our work makes the first attempt to use causal model to improve the performance of *link prediction* which is also an important graph learning task. Second, to make the attempt successful, our work presents a novel concept of "counterfactual link" and proposes a novel method CFLP that learns from both factual and counterfactual link existence. Third, the proposed method CFLP is flexible with the choice of treatment variables and is able to suggest good treatment choices prior to training via $\widehat{\text{ATE}}_{obs}$.

## C  DETAILS ON IMPLEMENTATION AND HYPERPARAMETERS

All the experiments in this work were conducted on a Linux server with Intel Xeon Gold 6130 Processor (16 Cores @2.1Ghz), 96 GB of RAM, and 4 RTX 2080Ti cards (11 GB of RAM each). Our method are implemented with `Python 3.8.5` with `PyTorch`. Source code is available in the supplementary materials. A list of used packages can be found in `requirements.txt`.

**Baseline Methods**  For baseline methods, we use official code packages from the authors for MV-GRL[4] (Hassani & Khasahmadi, 2020), SEAL[5] (Zhang & Chen, 2018), and LGLP[6] (Cai et al., 2021). We use a public implementation for VGAE[7] (Kipf & Welling, 2016b) and OGB implementations[8] for Node2Vec and baseline GNNs. For fair comparison, we set the size of node/link representations to be 256 of all methods.

**CFLP**  We use the Adam optimizer with a simple cyclical learning rate scheduler (Smith, 2017), in which the learning rate waves cyclically between the given learning rate ($lr$) and 1e-4 in every 70 epochs (50 warmup steps and 20 annealing steps). We implement the GNN encoders with `torch_geometric`[9] (Fey & Lenssen, 2019). Same with the baselines, we set the size of all hidden layers and node/link representations of CFLP as 256. The graph encoders all have three layers and JKNet has mean pooling for the final aggregation layer. The decoder is a 3-layer MLP with a hidden layer of size 64 and ELU as the nonlinearity. As the Euclidean distance used in Eq. (3) has a range of $[0, \infty)$, the value of $\gamma$ depends on the distribution of all-pair node embedding distances, which varies for different datasets. Therefore, we set the value of $\gamma$ as the $\gamma_{pct}$-percentile of all-pair node embedding distances. Commands for reproducing the experiments are included in `README.md`.

**Hyperparameter Searching Space**  We manually tune the following hyperparameters over range: $lr \in \{0.005, 0.01, 0.05, 0.1, 0.2\}$, $\alpha \in \{0.001, 0.01, 0.1, 1, 2\}$, $\beta \in \{0.001, 0.01, 0.1, 1, 2\}$, $\gamma_{pct} \in \{10, 20, 30\}$.

**Treatments**  For the graph clustering or community detection methods we used as treatments, we use the implementation from `scikit-network`[10] for Louvain (Blondel et al., 2008), SpecC (Ng et al., 2001), PropC (Raghavan et al., 2007), and Ward (Ward Jr, 1963). We used implementation of K-core (Bader & Hogue, 2003) from `networkx`[11] We used SBM (Karrer & Newman, 2011) from a public implementation by Funke & Becker (2019).[12] For CommN and Katz, we set $T_{i,j} = 1$ if the number of common neighbors or Katz index between $v_i$ and $v_j$ are greater or equal to 2 or 2 times the average of all Katz index values, respectively. For SpecC, we set the number of clusters as 16. For SBM, we set the number of communities as 16. These settings are fixed for all datasets.

---

[4] `https://github.com/kavehhassani/mvgrl`

[5] `https://github.com/facebookresearch/SEAL_OGB`

[6] `https://github.com/LeiCaiwsu/LGLP`

[7] `https://github.com/DaehanKim/vgae_pytorch`

[8] `https://github.com/snap-stanford/ogb/tree/master/examples/linkproppred/ddi`

[9] `https://pytorch-geometric.readthedocs.io/en/latest/`

[10] `https://scikit-network.readthedocs.io/`

[11] `https://networkx.org/documentation/`

[12] `https://github.com/funket/pysbm`

Table 6: Link prediction performances measured by Hits@50. Best performance and best baseline performance are marked with bold and underline, respectively.

|  | CORA | CITESEER | PUBMED | FACEBOOK | OGB-DDI |
|---|---|---|---|---|---|
| Node2Vec | 63.64±0.76 | 54.57±1.40 | 50.73±1.10 | 43.91±1.03 | 24.34±1.67 |
| MVGRL | 29.97±3.06 | 26.48±0.98 | 16.96±0.56 | 17.06±0.19 | 12.03±0.11 |
| VGAE | 60.36±2.71 | 54.68±3.15 | 41.98±0.31 | 51.36±0.93 | 23.00±1.66 |
| SEAL | 51.68±2.85 | 54.55±1.77 | 42.85±2.03 | 57.20±1.85 | 40.85±2.97 |
| LGLP | 71.43±0.75 | 69.98±0.16 | – | 56.22±0.49 | – |
| GCN | 64.93±1.62 | 63.38±1.73 | 39.20±6.47 | 69.90±0.65 | 73.70±3.99 |
| GSAGE | 63.18±3.39 | 61.71±2.43 | 54.81±2.67 | 62.53±4.24 | 86.83±3.85 |
| JKNet | 62.64±1.40 | 62.26±2.10 | 45.16±3.18 | 68.81±1.76 | 91.48±2.41 |
| Our proposed CFLP with different graph encoders | | | | | |
| CFLP w/ GCN | 72.61±0.92 | 69.85±1.11 | 55.00±1.95 | 70.47±0.77 | 62.47±1.53 |
| CFLP w/ GSAGE | 73.25±0.94 | 64.75±2.27 | 58.16±1.40 | 63.89±2.08 | 83.32±3.61 |
| CFLP w/ JKNet | **75.49**±1.54 | **77.01**±1.92 | **62.80**±0.79 | **71.41**±0.61 | **93.07**±1.14 |

Table 7: Link prediction performances measured by Average Precision (AP). Best performance and best baseline performance are marked with bold and underline, respectively.

|  | CORA | CITESEER | PUBMED | FACEBOOK | OGB-DDI |
|---|---|---|---|---|---|
| Node2Vec | 88.53±0.42 | 84.42±0.48 | 87.15±0.12 | 99.07±0.02 | 98.39±0.04 |
| MVGRL | 76.47±3.07 | 67.40±0.52 | 82.00±0.97 | 82.37±0.35 | 81.12±1.77 |
| VGAE | 89.89±0.50 | 86.97±0.78 | 95.97±0.16 | 98.60±0.04 | 95.28±0.11 |
| SEAL | 89.08±0.57 | 88.55±0.32 | 96.33±0.28 | **99.51**±0.03 | 98.39±0.21 |
| LGLP | 93.05±0.03 | 91.62±0.09 | – | 98.62±0.01 | – |
| GCN | 91.42±0.45 | 90.87±0.52 | 96.19±0.88 | 99.42±0.02 | 99.86±0.03 |
| GSAGE | 91.52±0.46 | 89.43±1.15 | 96.93±0.11 | 99.27±0.06 | 99.93±0.01 |
| JKNet | 90.50±0.22 | 90.42±1.34 | 96.56±0.31 | 99.41±0.02 | 99.95±0.01 |
| Our proposed CFLP with different graph encoders | | | | | |
| CFLP w/ GCN | 93.77±0.49 | 91.84±0.20 | 97.16±0.08 | 99.40±0.01 | 99.60±0.03 |
| CFLP w/ GSAGE | 93.55±0.49 | 90.80±0.87 | 97.10±0.08 | 99.29±0.06 | 99.88±0.04 |
| CFLP w/ JKNet | **94.24**±0.28 | **93.92**±0.41 | **97.69**±0.13 | 99.35±0.02 | **99.96**±0.01 |

# D  ADDITIONAL EXPERIMENTAL RESULTS AND DISCUSSIONS

**Link Prediction**  Tables 6 and 7 show the link prediction performance of Hits@50 and Average Precision (AP) by all methods. LGLP on PUBMED and OGB-DDI are missing due to the out of memory error when running the code package from the authors. Similar to the results in Tables 1 and 2, we observe that our CFLP on different graph encoders achieve similar or better performances compared with baselines, with the only exception of AP on FACEBOOK where most methods have close-to-perfect AP. From Tables 1, 2, 6 and 7, we observe that CFLP achieves improvement over all GNN architectures (averaged across datasets). Specifically, CFLP improves 25.6% (GCN), 12.0% (GSAGE), and 36.3% (JKNet) on Hits@20, 9.6% (GCN), 5.0% (GSAGE), and 17.8% (JKNet) on Hits@50, 5.6% (GCN), 1.6% (GSAGE), and 1.9% (JKNet) on AUC, and 0.8% (GCN), 0.8% (GSAGE), and 1.8% (JKNet) on AP. We note that CFLP with JKNet almost consistently achieves the best performance and outperforms baselines significantly on Hits@50. Specifically, compared with the best baseline, CFLP improves relatively by 6.8% and 0.9% on Hits@50 and AP, respectively.

**Ablation Study on Losses**  For the ablative studies of $\mathcal{L}_{CF}$ (Eq. (9)) and $\mathcal{L}_{disc}$ (Eq. (10)), we show their effect by removing them from the integrated loss function (Eq. (11)). Table 8 shows the results of CFLP on CORA and CITESEER under different settings ($\alpha = 0$, $\beta = 0$, $\alpha = \beta = 0$, and original setting). We observe that CFLP in the original setting achieves the best performance. The performance drops significantly when having $\alpha = 0$, i.e., not using any counterfactual data during training. We note that having $\beta = 0$, i.e., not using the discrepancy loss, also lowers the performance. Therefore, both $\mathcal{L}_{CF}$ and $\mathcal{L}_{disc}$ are essential for improving the link prediction performance.

Table 8: Link prediction performance of CFLP (w/ JKNet) on CORA and CITESEER when removing $\mathcal{L}_{CF}$ or $\mathcal{L}_{disc}$ or both versus normal setting.

| | CORA | | CITESEER | |
| --- | --- | --- | --- | --- |
| | Hits@20 | AUC | Hits@20 | AUC |
| CFLP ($\alpha = 0$) | 58.58±0.23 | 89.16±0.93 | 65.49±2.18 | 91.01±0.64 |
| CFLP ($\beta = 0$) | 62.27±0.84 | 92.96±0.34 | 66.92±1.84 | 91.98±0.17 |
| CFLP ($\alpha = \beta = 0$) | 58.52±0.83 | 88.79±0.28 | 64.69±3.25 | 90.61±0.64 |
| CFLP | **65.57**±1.05 | **93.05**±0.24 | **68.09**±1.49 | **92.12**±0.47 |

Table 9: Link prediction performance of CFLP (w/ JKNet) on CORA and CITESEER with node embeddings ($\tilde{\mathbf{X}}$) learned from different methods.

| | CORA | | CITESEER | | OGB-DDI | |
| --- | --- | --- | --- | --- | --- | --- |
| | Hits@20 | AUC | Hits@20 | AUC | Hits@20 | AUC |
| (MVGRL) | 65.57±1.05 | 93.05±0.24 | 68.09±1.49 | 92.12±0.47 | 86.08±1.98 | 99.94±0.01 |
| (GRACE) | 62.54±1.41 | 92.28±0.69 | 68.68±1.75 | 93.80±0.36 | 82.30±3.32 | 99.93±0.01 |
| (DGI) | 61.04±1.52 | 92.99±0.49 | 72.17±1.08 | 93.34±0.51 | 85.61±1.66 | 99.94±0.01 |

**Ablation Study on Node Embedding $\tilde{\mathbf{X}}$**   As the node embedding $\tilde{\mathbf{X}}$ is used in the early step of CFLP for finding the counterfactual links, the quality of $\tilde{\mathbf{X}}$ may affect the later learning process. Therefore, we also evaluate CFLP with different state-of-the-art unsupervised graph representation learning methods: MVGRL (Hassani & Khasahmadi, 2020), DGI (Velickovic et al., 2019), and GRACE (Zhu et al., 2020). Table 9 shows the link prediction performance of CFLP (w/ JKNet) on CORA and CITESEER with different node embeddings. We observe that the choice of the method for learning $\tilde{\mathbf{X}}$ does have an impact on the later learning process as well as the link prediction performance. Nevertheless, Table 9 shows CFLP's advantage can be consistently observed with different choices of methods for learning $\tilde{\mathbf{X}}$, as CFLP with $\tilde{\mathbf{X}}$ learned from all three methods showed promising link prediction performance.

**Sensitivity Analysis of $\alpha$ and $\beta$**   Figure 4 shows the AUC performance of CFLP on CORA with different combinations of $\alpha$ and $\beta$. We observe that the performance is the poorest when $\alpha = \beta = 0$ and gradually improves and gets stable as $\alpha$ and $\beta$ increase, showing that CFLP is generally robust to the hyperparameters $\alpha$ and $\beta$, and the optimal values are easy to locate.

**Sensitivity Analysis of $\gamma$**   Figure 5 shows the Hits@20 and AUC performance on link prediction of CFLP (with JKNet) on CORA and CITESEER with different treatments and $\gamma_{pct}$. We observe that the performance is generally good when $10 \leq \gamma_{pct} \leq 20$ and gradually get worse when the value of $\gamma_{pct}$ is too small or too large, showing that CFLP is robust to $\gamma$ and the optimal $\gamma$ is easy to find.

**Sensitivity to Noisy Data**   We note that robustness w.r.t. noisy data is not within our claim of technical contributions. Nevertheless, CFLP is not more vulnerable than other link prediction baselines. We conduct experiments with random attacks on the Cora dataset (randomly removing links and adding noisy links). Table 10 shows the AUC performances of our proposed CFLP (w/ JKNet) compared to the strongest baseline methods under different levels of random attacks (0%, 2%, 5%, and 10%). We can observe that as the attack strength goes up, the link prediction performance of all methods go down. We also note that our proposed CFLP still outperforms the baseline methods.

**Generalization to Graphs with Weighted Edges**   As our proposed CFLP uses GNN as the graph encoder and GNNs are usually able to take weighted graph as input (e.g., the adjacency matrix $\mathbf{A}$ for GCN can be weighted), the model should be able to handle weighted graphs as given. Note the link prediction losses (Eqs. (8) and (9)) need to be slightly modified considering the task. When the task is to predict the link existence, the label adjacency matrix used in Eqs. (8) and (9) must be of binary values. When the task is to predict the link weights, the BCE loss functions (Eqs. (8) and (9)) need to be changed to regression loss functions such as MSE.

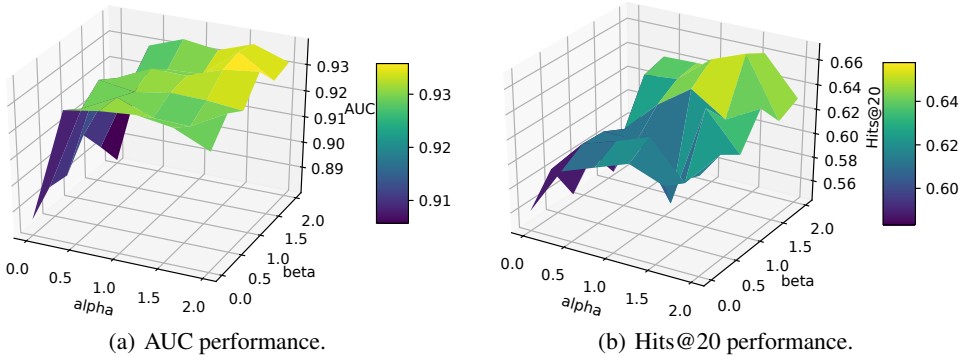

(a) AUC performance.

(b) Hits@20 performance.

Figure 4: Performance of CFLP on CORA w.r.t different combinations of $\alpha$ and $\beta$.

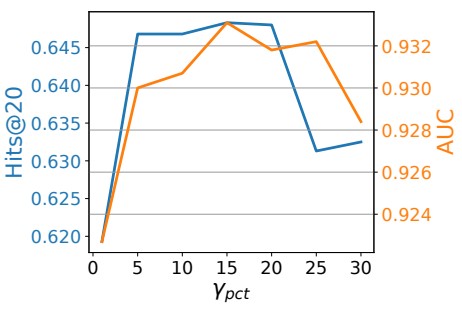

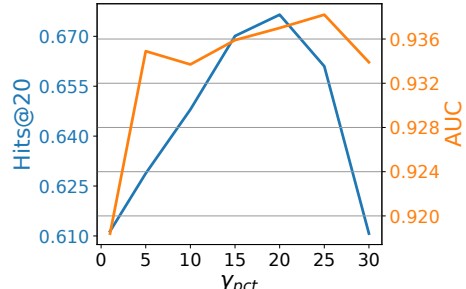

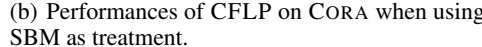

(a) Performances of CFLP on CORA when using K-core as treatment.

(b) Performances of CFLP on CORA when using SBM as treatment.

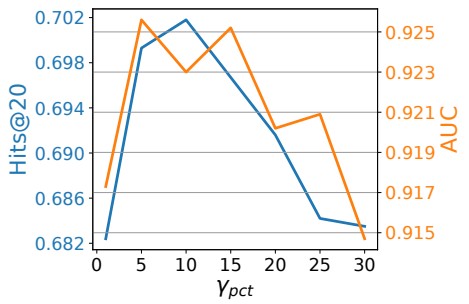

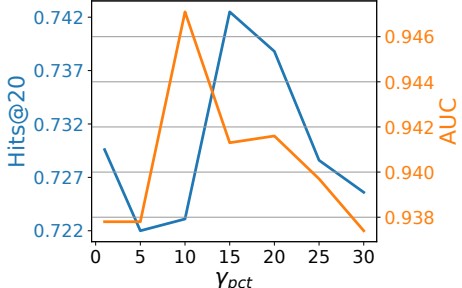

(c) Performances of CFLP on CITESEER when using K-core as treatment.

(d) Performances of CFLP on CITESEER when using SBM as treatment.

Figure 5: Hits@20 and AUC performances of CFLP (w/ JKNet) on CORA and CITESEER with different treatments w.r.t. different $\gamma_{pct}$ value.

Table 10: Link prediction performances measured by AUC on CORA when the graph is randomly perturbed at different levels. Best performances are marked with bold.

|  | 0% | 2% | 5% | 10% |
|---|---|---|---|---|
| SEAL | 92.55±0.50 | 87.81±1.37 | 87.90±0.79 | 87.64±0.89 |
| LGLP | 91.30±0.05 | 90.70±0.15 | 90.38±0.17 | 88.61±0.15 |
| JKNet | 89.05±0.67 | 88.85±0.75 | 88.14±0.59 | 87.64±0.89 |
| CFLP w/ JKNet | **93.05**±0.24 | **92.93**±0.17 | **92.77**±0.18 | **91.58**±0.23 |

