# OpenReview forum: "Counterfactual Graph Learning for Link Prediction"
_ICLR.cc/2022/Conference — ICLR 2022 Submitted_

### Official Review · Reviewer_LKEp · 2021-10-26

**Correctness:** 3
**Technical Novelty And Significance:** 3
**Empirical Novelty And Significance:** 3
**Recommendation:** 5
**Confidence:** 4

**Details Of Ethics Concerns:**

Not observed

**Main Review:**

For experiment, I would recommend using more from OGB. I am curious as OGB has many ling prediction dataset, why authors only use one of them? In addition, OGB-DDI authors model should have ranked second best if author submit the results, I wonder why authors do not submit the results in the official leaderboard? I strongly recommending using official leaderboard to illustrate the performance.

CFLP w/ JKNet consistently achieves the best performance across all dataset, while JKNet itself is only very outstanding. Is there any insight on the impact of CFLP for different architecture?

For the node representation, authors used MVGRL. Is there any ablation study on the impact of the node embedding? This is the early step and if embedding quality is low, the error can propagate through the learning process.

Also authors mentioned the embedding is learnt from the observed graph, so my understanding the link in validation/testing set is removed during embedding learning?

In page 4, "That is, we want to find the nearest neighbor with the opposite treatment for each observed node pairs and use the nearest neighbor’s outcome as a counterfactual link." I would recommend adding reference to some matching based methods in causal inference.

In page 4, authors first used d for a distance function in (2) between two node pairs, while later used similar d for a distance function for two nodes. This is confusing and I would recommend using another symbol.

For RELATED WORK, it seems not clear how some literatures (especially in the causal inference section) are relevant to this work.

**Summary Of The Paper:**

In this work, authors presented a counterfactual graph learning method for link prediction (CFLP), where authors introduced the idea of counterfactual prediction to improve link prediction on graphs.

Authors demonstrate the model performance on link prediction tasks and achieved promising results. Such results shed insights that a good use of causal models (even basic ones) can greatly improve the performance of (graph) machine learning tasks.

**Summary Of The Review:**

From the quality and novelty perspective, the proposed method is interesting and the experiment is thorough. However, my major concern is the causal model in the paper. I am not very convinced with the causal model with 'node similarity' as a treatment, and 'link' as outcome. The proposed approach is more like a data augmentation, where similar pairs are found with different community-relation to enrich the training set. It would be helpful to spend more effort to illustrate why such causal graph design makes sense.

In addition, the experiment results are exciting, but I didn't observe any submission in the open benchmark website. I highly recommend make a submission for a fair comparison, or explain why the submission has not done yet.

---

> ### Author Response · Authors · 2021-11-17
> **Response to reviewer LKEp (part 2/2)**
>
> 6. **Is there any ablation study on the impact of the node embedding?**
>
> This is a very good point and we agree that the quality of the embedding would affect the learning process. We conducted an ablation study on the impact of $\tilde{\mathbf{X}}$ by evaluating our proposed CFLP with different state-of-the-art unsupervised graph representation learning methods: MVGRL, DGI, and GRACE. The table below shows the link prediction performance of CFLP (w/ JKNet) on Cora and CiteSeer with different node embeddings. We observe that the choice of the method for learning $\tilde{\mathbf{X}}$ does have an impact on the later learning process as well as the link prediction performance. Nevertheless, the results show that CFLP’s advantage can be consistently observed with different choices of methods of learning $\tilde{\mathbf{X}}$, as CFLP with $\tilde{\mathbf{X}}$ learned from all three methods showed promising link prediction performance. \
> We have included the table along with the observations in Table 9 and Appendix C.
>
> |              |   Cora  |       | CiteSeer |       |  OGB-ddi |       |
> |--------------|:-------:|:-----:|:--------:|:-----:|:--------:|:-----:|
> |              | Hits@20 |  AUC  |  Hits@20 |  AUC  |  Hits@20 |  AUC  |
> | CFLP (MVGRL) |  65.57  | 93.05 |   68.09  | 92.12 |   86.08  | 99.94 |
> | CFLP (GRACE) |  62.54  | 92.28 |   68.68  | 93.80 |   82.30  | 99.93 |
> | CFLP (DGI)   |  61.04  | 92.99 |   72.17  | 93.34 |   85.61  | 99.94 |
>
> 7. **Authors mentioned that the embedding is learnt from the observed graph, so my understanding is that the link in validation/testing set is removed during embedding learning?**
>
> Yes, all validation and testing links are masked off during embedding learning to ensure fair comparison. \
> We added clarification in the paragraph above Eq. (3) in Section 3.2.
>
> 8. **Add reference to matching based methods in causal inference.**
>
> Thanks for pointing this out. We have edited the sentence with proper citations.
>
> 9. **Use another symbol for $d$ in Eq. (2).**
>
> Thanks for pointing this out. We have changed it to another symbol $h$ in text.

---

> ### Author Response · Authors · 2021-11-17
> **Response to reviewer LKEp (part 1/2)**
>
> We appreciate your careful reading and insightful comments. Following are the detailed responses regarding your concerns (re-ordered by putting the main concerns in summary at first).
>
> 1. **My major concern is the causal model in the paper. I am not very convinced with the causal model with “node similarity” as a treatment, and “link” as outcome.**
>
> As defined in the introduction and Section 3.1, our proposed work used “global structural information” (e.g., whether two nodes belong to the same community) as treatment and “link existence” as outcome. As argued in the second paragraph of Introduction, conventional link prediction models would be confused by non-essential factors such as global structural roles. Take the example we used in the Introduction: Alice and Adam live in the same neighborhood and they are close friends. The correlation between neighborhood belonging and friend closeness in the data could be too strong for the link prediction model to discover the essential factors (causes) of the friendship such as common interests or family relationship. So with our model, if Helen and Bob were not in the same neighborhood but shared many common interests, the likeliness of them being friends could still be captured and predicted. Therefore, we utilize the basic causal model (introduced in Section 3.1) to enhance the graph representation learning for link prediction.
>
> 2. **The proposed approach is more like a data augmentation, where similar pairs are found with different community-relation to enrich the training set. It would be helpful to spend more effort to illustrate why such causal graph design makes sense.**
>
> Thank you for the suggestion. We agree that CFLP can be a new type of approach for graph data augmentation, because the link prediction model learns from both observed links and counterfactual links. This novel approach demonstrated its effectiveness on the task of link prediction with significant empirical improvements. It is absolutely interesting to study the relationship between graph data augmentation and causal inference on graph learning. We take it as a future work and encourage the graph learning community to take a look at it as well. \
> We have added this as part of the future work in Section 6.
>
> 3. **I wonder why authors do not submit the results in the official OGB leaderboard?**
>
> Thank you for the suggestion. We did not submit our results to the OGB leaderboard because we were concerned about violating the double blind reviewing policy. It is because submitting to the OGB leaderboard requires a contact name, paper link, and a public repository. We recently took a look at the Author’s Guide of ICLR. It says: “It is ok to report the results on the leaderboard of a challenge. The authors can include the ranking and the name of the challenge. The reviewers will be advised to not intentionally search the authors by examining the leaderboard.” So we are submitting our results to the leaderboard while we suggest reviewers to not intentionally search the authors’ names. Our code has been public for reviewers to reproduce the results (as in the paper and supplementary material).
>
> 4. **Why authors only use one of OGB datasets?**
>
> In the experiments, we use five benchmark datasets from 3 different domain/sources: citation networks (Cora, CiteSeer, and PubMed), social network (Facebook), and drug-drug interaction network (OGB-ddi). We also note that the three citation networks are the most used public datasets for GNN-related literature. We think the existing experimental results should well demonstrate that our proposed CFLP achieves state-of-the-art performance for link prediction.
>
> 5. **Is there any insight on the impact of CFLP for different architecture?**
>
> Thank you for pointing this out. From Tables 2 and 3 we observe that our proposed CFLP improves the performance on link prediction over all GNN architectures (averaged across datasets). More specifically, CFLP improves 25.6% (GCN), 12.0% (GSAGE), and 36.3% (JKNet) on Hits@20 and 5.6% (GCN), 1.6% (GSAGE), and 1.9% (JKNet) on AUC. CFLP benefits from learning with both observed link existence ($\mathbf{A}$) and our defined counterfactual links ($\mathbf{A}^{CF}$) and achieves outstanding improvements over all GNN architectures. \
> We added the above observations in Appendix C.

---

> ### Author Response · Authors · 2021-11-22
> **Looking forward to your feedback**
>
> Dear Reviewer LKEp,
>
> Thank you for your valuable suggestions again. We have responded to your initial comments. We have submitted our results to the OGB leaderboard on Nov. 17th. It should appear on the leaderboard soon accroding to their website: "The results will be posted after we check the model validity (expect to take about about a week)." \
> We are looking forward to your feedback and are glad to answer your further questions.
>
> Best,
> Authors

---

> ### Author Response · Authors · 2021-11-28
> **Results posted on the OGB leaderboard**
>
> Dear Reviewer LKEp,
>
> Hope you had a great Thanksgiving holiday. The OGB team has finished validating our results and posted our results on OGB-ddi on their official leaderboard (ranked as 2nd best). Nevertheless, according to the Author Guide of ICLR, we still suggest reviewers to not intentionally search the authors’ names.
>
> Best,\
> Authors

---

### Official Review · Reviewer_Q6cy · 2021-11-02

**Correctness:** 3
**Technical Novelty And Significance:** 4
**Empirical Novelty And Significance:** Not applicable
**Recommendation:** 8
**Confidence:** 4

**Main Review:**

Strengths:

- Highly creative and novel approach, to the best of my knowledge. The closest related work I have seen is to add and remove edges to a graph to try to improve link prediction accuracy, but that is very different to the counterfactual learning approach proposed here.
- Proposed approach is conceptually easy to understand and could lead to many variants in the future.
- Strong empirical performance compared to other recent GNNs.

Weaknesses:

- The proposed counterfactual learning formulation feels like a hammer looking for a nail. The way the treatment is defined seems highly unusual, and there is nothing really principled about it.
- The authors claim that the estimates of average treatment effect (ATE) are generally close to the observed ATE, but I don't see this at all from Tables 3 and 4. The estimated ATEs are close only for a few specific cases. I do agree with the second conclusion about the ranking of estimated ATE being useful to select the treatment, however.

Question regarding a claim from Page 5 section on Complexity:  "When $\gamma$ is set as a small value to obtain indeed similar node pairs, this step (Eq. (3)) uses constant time." Why is this the case? Also, what is $|\mathcal{E}|$? Is this the total number of edges in the graph?

Typo:

- Page 3, 4th paragraph: "Traditional causal inference methods hence statistical learning approaches". "hence" should probably be replaced with "use"

*After discussion period:*
I have lowered my score slightly following the discussions but continue to support the paper. Despite the unusual framing as counterfactual learning as compared to data augmentation, the empirical gains are highly impressive. I think it could inspire future research investigating the use of other structural properties of the graph as the "treatment".

**Summary Of The Paper:**

The authors propose an approach for link prediction using counterfactual learning. They define a "treatment" for a node pair as whether or not they belong in the same group (e.g. from running a graph clustering algorithm). They then find the most similar node pair with a different treatment and treat its outcome (whether there is a link) as the counterfactual of the outcome of the original node pair. To find the most similar node pair, they use a graph neural network (GNN) based encoder. They use two link decoders (both multi-layer Perceptrons) to predict both the actual and counterfactual outcomes. They demonstrate impressive improvements on link prediction accuracy on 5 real data sets.

**Summary Of The Review:**

The authors propose a highly creative approach to improve link prediction accuracy by an innovative definition of treatment for a node pair in a counterfactual learning set up. The proposed approach yields highly impressive empirical link prediction accuracy.

---

> ### Author Response · Authors · 2021-11-17
> **Response to reviewer Q6cy**
>
> We appreciate your careful reading and insightful comments. Following are the detailed responses regarding your concerns.
>
> 1. **The way the treatment is defined seems highly unusual, and there is nothing really principled about it.**
>
> Our proposed method uses graph structural properties (i.e.,  whether two nodes belong to the same community) as treatments. Such properties are kind of usual in the context of link prediction task, because community detection has been well studied in the past few decades. Communities have long been serving as one of the essential structural properties of graphs. Nevertheless, we note that other treatment choices are also worth further study. And theoretical analysis for the treatment choice could also be a valuable future research direction. \
> We have added this as part of the future work in Section 6.
>
> 2. **The authors claim that the estimates of average treatment effect (ATE) are generally close to the observed ATE, but I don't see this at all from Tables 3 and 4.**
>
> Thank you for your careful reading and pointing this out. We agree that the values of $\widehat{\text{ATE}}\_{est}$ and $\widehat{\text{ATE}}\_{obs}$ are not very close. Nevertheless, we note that their rankings are positively correlated with [Kendall’s coefficient](https://en.wikipedia.org/wiki/Kendall_rank_correlation_coefficient) of 0.67 and 0.57 for Cora and CiteSeer, respectively. Note that the range of this coefficient is [-1, 1]. \
> We have changed the text with more appropriate observations.
>
> 3. **Why Eq. (3) uses constant time when $\gamma$ is a small value.**
>
> In the implementation of CFLP, we pre-calculate the closest neighbors for each node (all-pair Euclidean distances and then argsort). Hence when finding the counterfactual links (Eq.(3)) with a small $\gamma$, we only need to compare with the first few nodes with smallest distances (similar to the algorithm for merging two sorted LinkedLists but stopping very early) for each node, whose time complexity can be viewed as constant. We also note that Eq.(3) would take linear time ($O(N)$) when the value of $\gamma$ is large.
>
> 4. **What is $|\mathcal{E}|$?**
>
> $|\mathcal{E}|$ is the number of edges in the observed graph, as $\mathcal{E}$ is defined as the set of observed links in Section 2.
>
> 5. **Typo.**
>
> Thank you for pointing it out. We have fixed it in text.

---

> ### Author Response · Authors · 2021-11-22
> **Looking forward to your feedback**
>
> Dear Reviewer Q6cy,
>
> Thank you for your encouraging comments again. We have responded to your initial comments. We are looking forward to your feedback and are glad to answer your further questions.
>
> Best,\
> Authors

---

> > ### Comment · Reviewer_Q6cy · 2021-11-29
> > **I still support the paper**
> >
> > Thanks to the authors for the detailed responses to all of the reviewers and area chair. I still support the paper and find the proposed approach to be highly innovative with strong empirical performance. I do find the way the contribution is presented as counterfactual graph learning to be potentially confusing and see why other reviewers have some concerns there.

---

> > > ### Author Response · Authors · 2021-11-29
> > > **Appreciation**
> > >
> > > We thank you for your continued support. Honestly, we are disappointed about losing two points when we have carefully addressed your concerns. Please let us know if you have any further questions to our previous response.
> > >
> > > About the potential confusion - If suggested, we are willing to change the title as "Learning from Counterfactual Graph for Link Prediction" to avoid the confusion. The key difference from the original title will be:
> > >
> > > "Counterfactual Graph Learning" $\rightarrow$ "Learning from Counterfactual Graph"
> > >
> > > In the title, "counterfactual graph" is the novel concept we propose; and "link prediction" is the research problem we study. So we believe this title will accurately reflect the idea and goal of this work. Please advise us if you have any further comments.
> > >
> > > Thank you!

---

### Official Review · Reviewer_y1dx · 2021-11-02

**Correctness:** 3
**Technical Novelty And Significance:** 3
**Empirical Novelty And Significance:** 3
**Recommendation:** 6
**Confidence:** 5

**Main Review:**

strengths:
- novelty:
To me, the new perspective and the main idea are novel in the context of link prediction.
- correctness:
To my knowledge, the idea and technical contribution look sound.

weaknesses:
- No obvious weaknesses were found in this paper.
- It is not clear why T^{CF}, A^{CF} = T, A, otherwise in Eq.(4)
- It is not easy to digest "...ITE of neighborhood assignment as 1-1=0..." for someone without causality background!

additional questions:
- can you discuss how to generalize your method to continuous-valued network data in the supplement? say, A_ij \in [0,1], represents the connection strength between nodes i and j.
- if possible, can you discuss or even conduct additional experiments to demonstrate the robustness of your method to missing (false zeros) and noisy link data, which is prevalent in large-scale networks.
- except AUROC score, have you considered PR score to evaluate the link prediction performance as PR score is only sensitive to non-zero links?

typos:
- pp.1 only were -> are
- Algo 1 model inferencing -> inference
- many places causal model -> causal model(s)

**Summary Of The Paper:**

This paper targets an interesting question in network analysis: what are the main causes leading to the creation of a link between a pair of two nodes in a given network observations.

Most previous related models assume a latent structure underlying observed network data, and suppose the creation of each link is driven by the latent structure behind two nodes associated with that link. Hence, the main cause effect resulting in the link between two nodes, may not be truly reflected by the underlying community structure.

To address this concern, the paper resorts to a counterfactual learning framework by learning counterfactual links between the most similar node pairs with a different treatment. Experimental results demonstrate the improved link prediction by the counterfactual graph learning method compared with latent community based methods.

**Summary Of The Review:**

Overall, I vote for accepting this paper considering its novelty on solving link prediction from a new perspective of counterfactual learning.
Nonetheless, my background on causality does not allow to check all of its correctness in such a short time.

---

> ### Author Response · Authors · 2021-11-17
> **Response to reviewer y1dx**
>
> We appreciate your careful reading and insightful comments. Following are the detailed responses regarding your concerns.
>
> 1. **It is not clear why $\mathbf{T}^{CF}, \mathbf{A}^{CF} = \mathbf{T}, \mathbf{A}$, otherwise in Eq. (4).**
>
> As mentioned in the paragraph above Eq. (4), in the case that no node pair satisfies the conditions in Eq. (3) (i.e., there is no node pair with opposite treatment that is close enough to the target node pair), we do not assign any nearest neighbor to ensure all the counterfactual links are similar enough with their target node pairs in the feature space. \
> We have added an explanation in the text above Eq. (4).
>
> 2. **It is not easy to digest "...ITE of neighborhood assignment as 1-1=0..." for someone without causality background.**
>
> Thank you for pointing this out. We have added a sentence explaining the ITE as well as what ITE=0 indicates in the Introduction.
>
> 3. **Discuss how to generalize your method to continuous-valued network data.**
>
> As our proposed method uses GNN as the graph encoder and GNNs are usually able to take weighted graph as input (e.g., the adjacency matrix $\mathbf{A}$ for GCN can be weighted), the model should be able to handle weighted graphs as given. Note the link prediction losses (Eqs. (8) and (9)) need to be slightly modified considering the task. When the task is to predict the link existence, the label adjacency matrix used in Eqs. (8) and (9) must be of binary values. When the task is to predict the link weights, the BCE loss functions (Eqs. (8) and (9)) need to be changed to regression loss functions such as MSE. \
> We have included the above discussion in Appendix C.
>
> 4. **If possible, can you discuss or even conduct additional experiments to demonstrate the robustness of your method to missing (false zeros) and noisy link data.**
>
> Thank you for your suggestion. We would like to point out that robustness is not within our claim of technical contributions. Nevertheless, we note that our proposed CFLP is not more vulnerable than the basic GNNs or other baselines. We conducted experiments with random attacks on the Cora dataset (randomly removing links and adding noisy links). The following table shows the AUC performances of our proposed CFLP compared to the strongest baseline methods under different levels of random attacks (0%, 2%, 5%, and 10%). We can observe that as the attack strength goes up, the link prediction performances of all methods go down. We note that CFLP still outperforms the baseline methods. \
> We have included the table along with the observations in Table 10 and Appendix C.
>
> |                 |    0% |    2% |    5% |   10% |
> |-----------------|:-----:|:-----:|:-----:|:-----:|
> | SEAL            | 92.55 | 87.81 | 87.90 | 85.60 |
> | LGLP            | 91.30 | 90.70 | 90.38 | 88.61 |
> | JKNet           | 89.05 | 88.85 | 88.14 | 87.64 |
> | CFLP (w/ JKNet) | 93.05 | 92.93 | 92.77 | 91.58 |
>
> 5. **Except AUROC score, have you considered PR score to evaluate the link prediction performance?**
>
> In addition to the metrics AUROC and Hits@K, we reported [Averaged Precision (AP)](https://scikit-learn.org/stable/modules/generated/sklearn.metrics.average_precision_score.html) in Table 7 (Appendix C). AP score summarizes the PR curve.
>
> 6. **Typos.**
>
> Thank you for pointing them out. We have fixed the typos in text.

---

> ### Author Response · Authors · 2021-11-22
> **Looking forward to your feedback**
>
> Dear Reviewer y1dx,
>
> Thank you for your valuable suggestions again. We have responded to your initial comments. We are looking forward to your feedback and are glad to answer your further questions.
>
> Best,\
> Authors

---

### Official Review · Reviewer_WwN8 · 2021-11-08

**Correctness:** 3
**Technical Novelty And Significance:** 3
**Empirical Novelty And Significance:** 3
**Recommendation:** 5
**Confidence:** 4

**Main Review:**

Overall, I found this paper to be an interesting approach, but there are a few items that give me pause. Most importantly, it's not entirely clear what the causal estimand is that is being measured. The authors use membership to a community as the running treatment and consider the existence of a link between nodes as the outcome. However, it's not entirely clear to me that what the authors are ultimately doing is measuring a causal effect (the central task does not appear to be measuring a difference between observed and counterfactual networks). Instead, it seems that the proposed procedure is improving performance by enforcing a kind of invariance in representation between the observed network and the closest perturbed network. I find this to be a very interesting and compelling approach, but unfortunately it appears to be a bit obscured by the presentation.

A few more minor points:

* Sherman & Shpitser (Intervening on Network Ties, UAI 2019) should probably be cited given the relative similarity in task

* It would be helpful if the definition of ATE provided by equations 13 and 14 appeared earlier in the text. It would give a lot more clarity to the presentation to my eyes to have a more formal definition of the estimand provided during the problem setup.

* The authors define the ATE as the expectation of the ITE. The ITE is also an expectation, but conditional on X, whereas the ATE is an unconditional measure (after marginalizing over X).


**Summary Of The Paper:**

This work looks at the problem of link prediction from the lens of causal inference. In particular, the authors introduce counterfactual links, which are links which would have existed under a different treatment (with a running example being the neighborhood identifier). The authors use this definition to define a training procedure which builds on prior art in neural net based causal inference that uses an IPM penalty to ensure balance in representation between treatment and control outcomes. Experimental results are provided which show strong empirical performance for the proposed model.

**Summary Of The Review:**

Overall, I think this is an interesting approach, but unfortunately I think the framing of the paper as a causal estimation paper, rather than a paper which uses counterfactuals a mechanism to improve invariance and performance makes the contributions of the paper slightly obscured.

---

> ### Author Response · Authors · 2021-11-17
> **Response to reviewer WwN8**
>
> We appreciate your careful reading and insightful comments.
>
> **Regarding your main concern on the framing of our work (described in main review and summary), we clarify as follows:**
>
> The research focus of this paper is not on causal estimation or developing a new causal model. The central task of this work is ***link prediction*** (as a graph machine learning task), which is not a new measurement to causal effect. This can be found in Introduction (Section 1), in Problem Definition (Section 2), and even in the paper’s title (“Counterfactual Graph Learning for Link Prediction”). \
> To improve link prediction, or say, to improve graph machine learning models on this task, we leverage causal models to derive “counterfactual link” that is a novel concept we propose and use in this work. We may regard creating counterfactual links as “building the closest perturbed network” (as in your comments). The perturbed network is built upon the idea we presented in the Introduction. The proposed link prediction method CFLP learns from both observed links and the counterfactual links. CFLP delivers the state-of-the-art link prediction performance (Tables 1 and 2). To the best of our knowledge, this is the first attempt to advance graph learning / link prediction methods with discovery from causal models. We would like to share this idea and novel solution with the Graph Machine Learning community.
>
> Following are the detailed responses regarding your minor concerns.
>
> 1. **Missing citation.**
>
> Thank you for pointing this out, we have added the citation in Section 5.
>
> 2. **Position of Eqs. (12) and (13).**
>
>
>
> The definition of ITE was provided in Section 3.1 (Eq. (1)) and ATE is the expectation of ITE over all node pairs (also introduced in Section 3.1). However, the mathematical definitions of observed ATE ($\widehat{\text{ATE}}\_{obs}$) and estimated ATE ($\widehat{\text{ATE}}\_{est}$) need explanations to  many symbols that are introduced in Section 3. For example, the predicted adjacency matrices $\widehat{\mathbf{A}}$ and $\widehat{\mathbf{A}}^{CF}$ are introduced in Section 3.3 (Eqs. (6) and (7)). So, it would make readers more confused if we moved Eqs. (12) and (13) to an earlier position in the text. \
> Moreover, as this work focuses on graph learning for link prediction rather than causal inferencing, $\widehat{\text{ATE}}\_{obs}$ and $\widehat{\text{ATE}}\_{est}$ are for validating the learning ability of our proposed CFLP and helping CFLP to choose the best treatment (see the paragraph above Eq. (12)).
>
> 3. **The authors define the ATE as the expectation of the ITE. The ITE is also an expectation, but conditional on X, whereas the ATE is an unconditional measure (after marginalizing over X).**
>
> ATE is commonly defined as the expectation of ITE in counterfactual learning literature [1][2][3]. \
> We are wondering what the symbol $\mathbf{X}$ in this comment refers to. We guess it may refer to the node feature matrix (as defined in Section 2). In our definition (Eq. (1)), ITE can be calculated by representation vectors $\mathbf{Z}$, which is not an expectation of something conditional on the node feature matrix $\mathbf{X}$. As introduced in Section 3.1, we leverage causal models to improve the learning of the representation vectors $\mathbf{Z}$.
>
> [1] Learning representations for counterfactual inference, ICML 2016. \
> [2] Estimating individual treatment effect: generalization bounds and algorithms, ICML 2017. \
> [3] Counterfactual representation learning with balancing weights, AISTATS 2021.

---

> > ### Comment · Area_Chair_pFcN · 2021-11-17
> > **Please expand comparison**
> >
> > Just a quick note to authors: "Sherman & Shpitser (2020) modeled social networks with causal DAGs." is not a good description of the differences and similarities between Sherman & Shpitser and your work. Please expand it ASAP.

---

> > > ### Author Response · Authors · 2021-11-17
> > > **Response to AC**
> > >
> > > Many thanks for your message. We apologize for shrinking too much the sentence that discussed the relationship between Sherman & Shpitser (2020) and our work, due to the hard page limit in ICLR paper format policy. We have updated the sentence to carefully discuss the relationship. Here is the updated sentence:
> > > >“Sherman& Shpitser (2020) proposed a new concept in causal modeling, called “network intervention”, that is a type of structural intervention in network contexts. It studied the effect of link creation on network structure changes.”
> > >
> > > The research problem that Sherman& Shpitser (2020) studied is causal estimation by intervention. Our research problem is link prediction using graph machine learning. As given in the feedback to Reviewer WwN8, the definition of our research problem has been given and clarified in the paper’s title, Abstract, Introduction, Problem Definition, and the Method sections. We hope our clarification could help reviewers have a better understanding of the research focus and technical contributions of our work.

---

> ### Author Response · Authors · 2021-11-22
> **Looking forward to your feedback**
>
> Dear Reviewer WwN8,
>
> Thank you for your valuable suggestions again. We have responded to your initial comments. We are looking forward to your feedback and are glad to answer your further questions.
>
> Best,\
> Authors

---

> > ### Comment · Reviewer_WwN8 · 2021-11-30
> > **Thank you for your response.**
> >
> > Thank you for your response. As I said in my initial review. However, I think that there are still significant issues in the presentation that make it difficult to parse the contributions of the paper. I echo the comments of the area chair, and think that the paper would greatly benefit from a (substantial) revision that (a) more clearly defines and differentiates the causal and non-causal aspects of the task, (b) more clearly specifies the causal model that is being utilized. At this time, I am not changing my score, but do think that this has many elements of very high quality work.

---

### Author Response · Authors · 2021-11-17
**Summary of changes**

Dear reviewers and chairs,

We appreciate the reviewers' and AC’s insightful suggestions. Other than the detailed responses we replied to each reviewer. The summary of changes we made in the paper are as follows (all changes are highlighted with blue-colored text):

1. We fixed the typos, added the missing citation, and added proper explanations and clarifications according to the reviews.

2. We updated the analysis on the first observation on Tables 3 and 4 in the second paragraph of Section 4.2.

3. We added a discussion of future work in Section 6 on alternative treatment design, theoretical analysis of treatment choices, and relationship with graph data augmentation.

4. In Appendix D, we added:
    * detailed observations on the impact of our proposed CFLP on different GNN architectures;
    * ablation study on the choice of unsupervised graph representation learning methods for learning the node embeddings $\tilde{\mathbf{X}}$ (Table 9);
    * sensitivity analysis of our proposed CFLP to noisy (perturbed) graph data (Table 10);
    * discussion on the generalization of CFLP to weighted graphs.

---

### Decision · Program_Chairs · 2022-01-20

**Decision:**

Reject

**Comment:**

This work proposes an observational (not counterfactual) link prediction method based on clustering and data augmentation. The total score went down during the rebuttal phase. One of the challenges is that the title confused reviewers. The authors proposed other titles. But in the end all proposed titles had variations of "counterfactual graph", which is what makes the title confusing.

I think if the work had been more empirically-focused and stayed away from a causal model, it would have been accepted. The idea is ingenious and it is likely that it indeed gives the empirical improvement in observational models that the authors claim. The authors have very scant citations to relevant works in the intersection of causality and representation learning anyways. But if the causal model must stay for future submissions, the authors should: (a) more clearly define and differentiate the causal and non-causal aspects of the task (and properly contextualize the work), and (b) more clearly specifies the causal model that is being utilized (and be more mathematically rigorous in general).

Unfortunately, the original description of the causal model in the paper was both sloppy and incorrect. I had to ask a total number of five (5) long clarifying questions (visible to authors and reviewers only) in order to understand the causal assumptions. In my back-and-forth with the authors, the authors proposed a total of 7 different causal DAGs to explain their method. Some of these DAGs contradicted each other and most of the answers were vague. My first advice to the authors is to always be mathematically precise for the benefit of reviewers and readers. Causality requires very clear assumptions and precise notation. For instance, the variables in the causal DAG given in Figures 2 and 3 had little to do with the variables in the actual graph model. The causal DAG must have the variables $A_{ij}$, $T_{ij}$, $C_i$, $C_j$, $z_i$, $z_j$, for all $i,j \in V$. The final causal DAG proposed by the authors in the discussion was:
- $z_i \to A_{ij}$
- $z_j \to A_{ij}$
- $z_i \to T_{ij}$ (may not be needed)
- $z_j \to T_{ij}$ (may not be needed)
- $C_i \to T_{ij}$
- $C_j \to T_{ij}$
- $T_{ij} \to A_{ij}$

Regarding the nature of $z_i$ and $z_j$: They need to be formally described as structural node embeddings of the graph, otherwise the DAG is incorrect. After digging and reading some of the references the authors use, I came to the conclusion that the empirical method indeed relies on structural embeddings (and the authors later confirmed it). This should be made **very** clear in the SCM description.

This final causal model after discussion looks reasonable. And one challenge that remains, however, is sampling or MAP point estimates of the posterior $P(C_i,C_j|A)$. All standard clustering methods that can use this DAG rely on node embeddings $z_i, z_j$ being positional embeddings, not structural ones. So, how are we actually obtaining $P(C_i,C_j|A)$? The authors would need to prove that standard clustering methods can perform this task (I have my doubts this is even true). I fear this issue could again derail the paper in a future submission if the counterfactual justification is used again. One of the reviewers raised concerns about the quality of the embedding already.

The proposed method is observational and does not need a causal description. But if a causal description is provided, it must be correct and the method must be theoretically sound. The causality part of the paper needs a substantial revision, unfortunately. I suggest rejection.